# Adversarial Lipschitz Regularization

**Dávid Terjék**
Robert Bosch Kft.
Budapest, Hungary
`david.terjek@hu.bosch.com`

## Abstract

Generative adversarial networks (GANs) are one of the most popular approaches when it comes to training generative models, among which variants of Wasserstein GANs are considered superior to the standard GAN formulation in terms of learning stability and sample quality. However, Wasserstein GANs require the critic to be 1-Lipschitz, which is often enforced implicitly by penalizing the norm of its gradient, or by globally restricting its Lipschitz constant via weight normalization techniques. Training with a regularization term penalizing the violation of the Lipschitz constraint explicitly, instead of through the norm of the gradient, was found to be practically infeasible in most situations. Inspired by Virtual Adversarial Training, we propose a method called Adversarial Lipschitz Regularization, and show that using an explicit Lipschitz penalty is indeed viable and leads to competitive performance when applied to Wasserstein GANs, highlighting an important connection between Lipschitz regularization and adversarial training.

## 1 Introduction

In recent years, Generative adversarial networks (GANs) (Goodfellow et al., 2014) have been becoming the state-of-the-art in several generative modeling tasks, ranging from image generation (Karras et al., 2018) to imitation learning (Ho and Ermon, 2016). They are based on an idea of a two-player game, in which a discriminator tries to distinguish between real and generated data samples, while a generator tries to fool the discriminator, learning to produce realistic samples on the long run. Wasserstein GAN (WGAN) was proposed as a solution to the issues present in the original GAN formulation. Replacing the discriminator, WGAN trains a critic to approximate the Wasserstein distance between the real and generated distributions. This introduced a new challenge, since Wasserstein distance estimation requires the function space of the critic to only consist of 1-Lipschitz functions.

To enforce the Lipschitz constraint on the WGAN critic, Arjovsky et al. (2017) originally used weight clipping, which was soon replaced by the much more effective method of Gradient Penalty (GP) (Gulrajani et al., 2017), which consists of penalizing the deviation of the critic's gradient norm from 1 at certain input points. Since then, several variants of gradient norm penalization have been introduced (Petzka et al., 2018; Wei et al., 2018; Adler and Lunz, 2018; Zhou et al., 2019b).

Virtual Adversarial Training (VAT) (Miyato et al., 2019) is a semi-supervised learning method for improving robustness against local perturbations of the input. Using an iterative method based on power iteration, it approximates the adversarial direction corresponding to certain input points. Perturbing an input towards its adversarial direction changes the network's output the most.

Inspired by VAT, we propose a method called Adversarial Lipschitz Regularization (ALR), enabling the training of neural networks with regularization terms penalizing the violation of the Lipschitz constraint explicitly, instead of through the norm of the gradient. It provides means to generate a pair for each input point, for which the Lipschitz constraint is likely to be violated with high probability. In general, enforcing Lipschitz continuity of complex models can be useful for a lot of applications. In this work, we focus on applying ALR to Wasserstein GANs, as regularizing or constraining Lipschitz continuity has proven to have a high impact on training stability and reducing mode collapse. Source code to reproduce the presented experiments is available at https://github.com/dterjek/adversarial_lipschitz_regularization.

Our contributions are as follows:

- We propose Adversarial Lipschitz Regularization (ALR) and apply it to penalize the violation of the Lipschitz constraint directly, resulting in Adversarial Lipschitz Penalty (ALP).
- Applying ALP on the critic in WGAN (WGAN-ALP), we show state-of-the-art performance in terms of Inception Score and Fréchet Inception Distance among non-progressive growing methods trained on CIFAR-10, and competitive performance in the high-dimensional setting when applied to the critic in Progressive Growing GAN trained on CelebA-HQ.

## 2 BACKGROUND

### 2.1 WASSERSTEIN GENERATIVE ADVERSARIAL NETWORKS

Generative adversarial networks (GANs) provide generative modeling by a generator network $g$ that transforms samples of a low-dimensional latent space $Z$ into samples from the data space $X$, transporting mass from a fixed noise distribution $P_Z$ to the generated distribution $P_g$. The generator is trained simultaneously with another network $f$ called the discriminator, which is trained to distinguish between fake samples drawn from $P_g$ and real samples drawn from the real distribution $P_r$, which is often represented by a fixed dataset. This network provides the learning signal to the generator, which is trained to generate samples that the discriminator considers real. This iterative process implements the minimax game

$$\min_g \max_f \mathbb{E}_{x \sim P_r} \log(f(x)) + \mathbb{E}_{z \sim P_Z} \log(1 - f(g(z))) \tag{1}$$

played by the networks $f$ and $g$. This training procedure minimizes the approximate Jensen-Shannon divergence (JSD) between $P_r$ and $P_g$ (Goodfellow et al., 2014). However, during training these two distributions might differ strongly or even have non-overlapping supports, which might result in gradients received by the generator that are unstable or zero (Arjovsky and Bottou, 2017).

Wasserstein GAN (WGAN) (Arjovsky et al., 2017) was proposed as a solution to this instability. Originating from Optimal Transport theory (Villani, 2008), the Wasserstein metric provides a distance between probability distributions with much better theoretical and practical properties than the JSD. It provides a smooth optimizable distance even if the two distributions have non-overlapping supports, which is not the case for JSD. It raises a metric $d_X$ from the space $X$ of the supports of the probability distributions $P_1$ and $P_2$ to the space of the probability distributions itself. For these purposes, the Wasserstein-$p$ distance requires the probability distributions to be defined on a metric space and is defined as

$$W_p(P_1, P_2) = \left( \inf_{\pi \in \Pi(P_1, P_2)} \mathbb{E}_{(x_1, x_2) \sim \pi} d_X(x_1, x_2)^p \right)^{\frac{1}{p}}, \tag{2}$$

where $\Pi(P_1, P_2)$ is the set of distributions on the product space $X \times X$ whose marginals are $P_1$ and $P_2$, respectively. The optimal $\pi$ achieving the infimum in (2) is called the optimal coupling of $P_1$ and $P_2$, and is denoted by $\pi^*$. The case of $p = 1$ has an equivalent formulation

$$W_1(P_1, P_2) = \sup_{\|f\|_L \leq 1} \mathbb{E}_{x \sim P_1} f(x) - \mathbb{E}_{x \sim P_2} f(x), \tag{3}$$

called the Kantorovich-Rubinstein formula (Villani, 2008), where $f : X \to \mathbb{R}$ is called the potential function, $\|f\|_L \leq 1$ is the set of all functions that are 1-Lipschitz with respect to the ground metric $d_X$, and the Wasserstein-1 distance corresponds to the supremum over all 1-Lipschitz potential functions. The smallest Lipschitz constant for a real-valued function $f$ with the metric space $(X, d_X)$ as its domain is given by

$$\|f\|_L = \sup_{x, y \in X; x \neq y} \frac{|f(x) - f(y)|}{d_X(x, y)}. \tag{4}$$

Based on (3), the critic in WGAN (Arjovsky et al., 2017) implements an approximation of the Wasserstein-1 distance between $P_g$ and $P_r$. The minimax game played by the critic $f$ and the generator $g$ becomes

$$\min_g \max_{\|f\|_L \leq 1} \mathbb{E}_{z \sim P_Z} f(g(z)) - \mathbb{E}_{x \sim P_r} f(x), \tag{5}$$

a formulation that proved to be superior to the standard GAN in practice, with substantially more stable training behaviour and improved sample quality (Arjovsky et al., 2017), although recent GAN variants do not always use this objective (Brock et al., 2019). With WGAN, the challenge became effectively restricting the smallest Lipschitz constant of the critic $f$, sparking the birth of a plethora of Lipschitz regularization techniques for neural networks.

## 2.2 LIPSCHITZ FUNCTION APPROXIMATION

A general definition of the smallest Lipschitz constant of a function $f : X \to Y$ is

$$\|f\|_L = \sup_{x,y \in X; x \neq y} \frac{d_Y(f(x), f(y))}{d_X(x, y)}, \tag{6}$$

where the metric spaces $(X, d_X)$ and $(Y, d_Y)$ are the domain and codomain of the function $f$, respectively. The function $f$ is called Lipschitz continuous if there exists a real constant $K \geq 0$ for which $d_Y(f(x), f(y)) \leq K \cdot d_X(x, y)$ for any $x, y \in X$. Then, the function $f$ is also called K-Lipschitz. Theoretical properties of neural networks with low Lipschitz constants were explored in Oberman and Calder (2018), Bartlett (1998) and Drucker and LeCun (1992), showing that it induces better generalization.

Learning mappings with Lipschitz constraints became prevalent in the field of deep learning with the introduction of WGAN (Arjovsky et al., 2017). Enforcing the Lipschitz property on the critic was first done by clipping the weights of the network. This approach achieved superior results compared to the standard GAN formulation, but still sometimes yielded poor quality samples or even failed to converge. While clipping the weights enforces a global Lipschitz constant, it also reduces the function space, which might not include the optimal critic any more. Soon this method has been replaced by a softened one called Gradient Penalty (GP) (Gulrajani et al., 2017). Motivated by the fact that the optimal critic should have unit gradient norm on lines connecting the coupled points $(x_1, x_2) \sim \pi^*$ according to (2), they proposed a regularizer that enforces unit gradient norm along these lines, which not only enforces the Lipschitz constraint, but other properties of the optimal solution as well. However, $\pi^*$ is not known in practice, which is why Gulrajani et al. (2017) proposed to apply GP on samples of the induced distribution $P_i$, by interpolating samples from the marginals $P_1$ and $P_2$. The critic in the WGAN-GP formulation is regularized with the loss

$$\lambda \mathbb{E}_{x \sim P_i}(\|\nabla_x f(x)\|_2 - 1)^2 \tag{7}$$

where $P_i$ denotes the distribution of samples obtained by interpolating pairs of samples drawn from $P_r$ and $P_g$, and $\lambda$ is a hyperparameter acting as a Lagrange multiplier.

Theoretical arguments against GP were pointed out by Petzka et al. (2018) and Gemici et al. (2018), arguing that unit gradient norm on samples of the distribution $P_i$ is not valid, as the pairs of samples being interpolated are generally not from the optimal coupling $\pi^*$, and thus do not necessarily need to match gradient norm 1. Furthermore, they point out that differentiability assumptions of the optimal critic are not met. Therefore, the regularizing effect of GP might be too strong. As a solution, Petzka et al. (2018) suggested using a loss penalizing the violation of the Lipschitz constraint either explicitly with

$$\lambda \mathbb{E}_{x,y \sim P_\tau} \left( \frac{|f(x) - f(y)|}{\|x - y\|_2} - 1 \right)_+^2 \tag{8}$$

or implicitly with

$$\lambda \mathbb{E}_{x \sim P_\tau} (\|\nabla_x f(x)\|_2 - 1)_+^2 \tag{9}$$

where in both cases $(a)_+$ denotes $\max(0, a)$. The first method has only proved viable when used on toy datasets, and led to considerably worse results on relatively more complex datasets like CIFAR-10, which is why Petzka et al. (2018) used the second one, which they termed Lipschitz Penalty (LP). Compared to GP, this term only penalizes the gradient norm when it exceeds 1. As $P_\tau$ they evaluated the interpolation method described above, and also sampling random local perturbations of real and generated samples, but found no significant improvement compared to $P_i$. Wei et al. (2018) proposed dropout in the critic as a way for creating perturbed input pairs to evaluate the explicit Lipschitz penalty (8), which led to improvements, but still relied on using GP simultaneously.

A second family of Lipschitz regularization methods is based on weight normalization, restricting the Lipschitz constant of a network globally instead of only at points of the input space. One such

technique is called spectral normalization (SN) proposed in Miyato et al. (2018), which is a very efficient and simple method for enforcing a Lipschitz constraint with respect to the 2-norm on a per-layer basis, applicable to neural networks consisting of affine layers and K-Lipschitz activation functions. Gouk et al. (2018) proposed a similar approach, which can be used to enforce a Lipschitz constraint with respect to the 1-norm and $\infty$-norm in addition to the 2-norm, while also being compatible with batch normalization and dropout. Anil et al. (2019) argued that any Lipschitz-constrained neural network must preserve the norm of the gradient during backpropagation, and to this end proposed another weight normalization technique (showing that it compares favorably to SN, which is not gradient norm preserving), and an activation function based on sorting.

## 2.3 VIRTUAL ADVERSARIAL TRAINING

VAT (Miyato et al., 2019) is a semi-supervised learning method that is able to regularize networks to be robust to local adversarial perturbation. Virtual adversarial perturbation means perturbing input sample points in such a way that the change in the output of the network induced by the perturbation is maximal in terms of a distance between distributions. This defines a direction for each sample point called the virtual adversarial direction, in which the perturbation is performed. It is called virtual to make the distinction with the adversarial direction introduced in Goodfellow et al. (2015) clear, as VAT uses unlabeled data with virtual labels, assigned to the sample points by the network being trained. The regularization term of VAT is called Local Distributional Smoothness (LDS). It is defined as

$$L_{LDS} = D\left(p(y|x), p(y|x + r_{vadv})\right),\tag{10}$$

where $p$ is a conditional distribution implemented by a neural network, $D(p, p')$ is a divergence between two distributions $p$ and $p'$, for which Miyato et al. (2019) chose the Kullback-Leibler divergence (KLD), and

$$r_{vadv} = \arg\max_{\|r\|_2 \le \epsilon} D\left(p(y|x), p(y|x + r)\right)\tag{11}$$

is the virtual adversarial perturbation, where $\epsilon$ is a hyperparameter. VAT is defined as a training method with the regularizer (10) applied to labeled and unlabeled examples. An important detail is that (10) is minimized by keeping $p(y|x)$ fixed and optimizing $p(y|x + r_{vadv})$ to be close to it.

The adversarial perturbation is approximated by the power iteration $r_{vadv} \approx \epsilon r_k$, where

$$r_{i+1} \approx \frac{\nabla_r D\left(p(y|x), p(y|x + r)\right)\Big|_{r = \xi r_i}}{\left\|\nabla_r D\left(p(y|x), p(y|x + r)\right)\Big|_{r = \xi r_i}\right\|_2},\tag{12}$$

$r_0$ is a randomly sampled unit vector and $\xi$ is another hyperparameter. This iterative scheme is an approximation of the direction at $x$ that induces the greatest change in the output of $p$ in terms of the divergence $D$. Miyato et al. (2019) found that $k = 1$ iteration is sufficient in practical situations.

## 3 ADVERSARIAL LIPSCHITZ REGULARIZATION

Adler and Lunz (2018) argued that penalizing the norm of the gradient as in (9) is more effective than penalizing the Lipschitz quotient directly as in (8), as the former penalizes the slope of $f$ in all spatial directions around $x$, unlike the latter, which does so only along $(x - y)$. We hypothesize that using the explicit Lipschitz penalty in itself is insufficient because if one takes pairs of samples $x, y$ randomly from $P_r$, $P_g$ or $P_i$ (or just one sample and generates a pair for it with random perturbation), the violation of the Lipschitz penalty evaluated at these sample pairs will be far from its maximum, hence a more sophisticated strategy for sampling pairs is required. As we will show, a carefully chosen sampling strategy can in fact make the explicit penalty favorable over the implicit one.

Consider the network $f$ as a mapping from the metric space $(X, d_X)$ to the metric space $(Y, d_Y)$. Let us rewrite (6) with $y = x + r$ to get

$$\|f\|_L = \sup_{x, x+r \in X; 0 < d_X(x, x+r)} \frac{d_Y(f(x), f(x+r))}{d_X(x, x+r)}.\tag{13}$$

A given mapping $f$ is K-Lipschitz if and only if for any given $x \in X$, taking the supremum over $r$ in (13) results in a value $K$ or smaller. Assuming that this supremum is always achieved for some $r$, we

can define a notion of adversarial perturbation with respect to the Lipschitz continuity for a given $x \in X$ as

$$r_{adv} = \underset{x+r \in X; 0 < d_X(x,x+r)}{\arg\max} \frac{d_Y(f(x), f(x+r))}{d_X(x, x+r)}, \qquad (14)$$

and the corresponding maximal violation of the K-Lipschitz constraint as

$$L_{ALP} = \left( \frac{d_Y(f(x), f(x+r_{adv}))}{d_X(x, x+r_{adv})} - K \right)_+ . \qquad (15)$$

We define Adversarial Lipschitz Regularization (ALR) as the method of adding (15) as a regularization term to the training objective that penalizes the violation of the Lipschitz constraint evaluated at sample pairs obtained by adversarial perturbation. We call this term Adversarial Lipschitz Penalty (ALP).

To put it in words, ALP measures the deviation of $f$ from being K-Lipschitz evaluated at pairs of sample points where one is the adversarial perturbation of the other. If added to the training objective, it makes the learned mapping approximately K-Lipschitz around the sample points it is applied at. We found that in the case of the WGAN critic it is best to minimize (15) without keeping $f(x)$ fixed. See Appendix A.1 for the semi-supervised case and Appendix A.2 for how VAT can be seen as a special case of Lipschitz regularization.

## 3.1 Approximation of $r_{adv}$

In general, computing the adversarial perturbation (14) is a nonlinear optimization problem. A crude and cheap approximation is $r_{adv} \approx \epsilon r_k$, where

$$r_{i+1} \approx \frac{\nabla_r d_Y\left(f(x), f(x+r)\right)\Big|_{r=\xi r_i}}{\left\|\nabla_r d_Y\left(f(x), f(x+r)\right)\Big|_{r=\xi r_i}\right\|_2}, \qquad (16)$$

is the approximated adversarial direction with $r_0$ being a randomly sampled unit vector. The derivation of this formula is essentially the same as the one described in Miyato et al. (2019), but is included in Appendix A.3 for completeness. Unlike in VAT, we do not fix $\epsilon$, but draw it randomly from a predefined distribution $P_\epsilon$ over $\mathbb{R}_+$ to apply the penalty at different scales.

Theoretically, ALR can be used with all kinds of metrics $d_X$ and $d_Y$, and any kind of model $f$, but the approximation of $r_{adv}$ imposes a practical restriction. It approximates the adversarial perturbation of $x$ as a translation with length $\epsilon$ with respect to the 2-norm in the adversarial direction, which is only a perfect approximation if the ratio in (15) is constant for any $\epsilon > 0$. This idealized setting is hardly ever the case, which is why we see the search for other approximation schemes as an important future direction. There is a large number of methods for generating adversarial examples besides the one proposed in VAT (Shafahi et al., 2019; Wong et al., 2019; Khrulkov and Oseledets, 2018), which could possibly be combined with ALR either to improve the approximation performance or to make it possible with new kinds of metrics. The latter is important since one of the strengths of the Wasserstein distance is that it can be defined with any metric $d_X$, a fact that Adler and Lunz (2018) and Dukler et al. (2019) built on by extending GP to work with metrics other than the Euclidean distance. Adler and Lunz (2018) emphasized the fact that through explicit Lipschitz penalties one could extend WGANs to more general metric spaces as well.

## 3.2 Hyperparmeters

In practice, one adds the Monte Carlo approximation of the expectation (averaged over a minibatch of samples) of either (15) or the square of (15) (or both) to the training objective, multiplied by a Lagrange multiplier $\lambda$. While VAT adds the expectation of (10) to the training objective, for WGAN we have added the square of the expectation of (15). To train the Progressive GAN, we have added both the expectation and its square. In the semi-supervised setting, we added only the expectation similarly to VAT. We have found these choices to work best in these scenarios, but a principled answer to this question is beyond the scope of this paper. The target Lipschitz constant $K$ can be tuned by hand, or in the presence of labeled data it is possible to calculate the Lipschitz constant of the dataset

(Oberman and Calder, 2018). The hyperparameters of the approximation scheme are $k$, $\xi$ and those of $P_\epsilon$.

Choosing the right hyperparameters can be done by monitoring the number of adversarial perturbations found by the algorithm for which the Lipschitz constraint is violated (and hence contribute a nonzero value to the expectation of (15)), and tuning the hyperparameters in order to keep this number balanced between its maximum (which is the minibatch size) and its minimum (which is 0). If it is too high, it means that either $K$ is too small and should be increased, or the regularization effect is too weak, so one should increase $\lambda$. If it is too low, then either the regularization effect is too strong, or ALR is parametrized in a way that it cannot find Lipschitz constraint violations efficiently. In the former case, one should decrease $\lambda$. In the latter, one should either decrease $K$, tune the parameters of $P_\epsilon$, or increase the number of power iterations $k$ for the price of increased runtime. We have not observed any significant effect when changing the value of $\xi$ in any of the tasks considered.

### 3.3 Comparison with other Lipschitz regularization techniques

In terms of efficiency when applied to WGANs, ALR compares favorably to the implicit methods penalizing the gradient norm, and to weight normalization techniques as well, as demonstrated in the experiments section. See Appendix A.4 for a showcase of the differences between weight normalization methods, implicit penalty methods and explicit penalty methods, represented by SN, LP and ALR, respectively. The key takeaways are that

- penalty methods result in a softer regularization effect than SN,
- ALR is preferable when the regularized network contains batch normalization (BN) layers, and
- ALR gives more control over the regularization effect, which also means there are more hyperparameters to tune.

The performance of ALR mostly depends on the speed of the approximation of $r_{adv}$. The current method requires 1 step of backpropagation for each power iteration step, which means that running time will be similar to that of LP and GP with $k = 1$. SN is much cheaper computationally than each penalty method, although we believe ALR has the potential to become relatively cheap as well by adopting new techniques for obtaining adversarial examples (Shafahi et al., 2019).

## 4 WGAN-ALP

We specialize the ALP formula (15) with $f$ being the critic, $d_X(x, y) = \|x - y\|_2$, $d_Y(x, y) = |x - y|$ and $K = 1$, and apply it to the WGAN objective to arrive at a version with the explicit penalty, which uses adversarial perturbations as a sampling strategy. It is formulated as

$$\mathbb{E}_{z \sim P_Z} f(g(z)) - \mathbb{E}_{x \sim P_r} f(x) + \lambda \mathbb{E}_{x \sim P_{r,g}} \left( \frac{|f(x) - f(x + r_{adv})|}{\|r_{adv}\|_2} - 1 \right)_+^2, \qquad (17)$$

where $P_{r,g}$ is a combination of the real and generated distributions (meaning that a sample $x$ can come from both), $\lambda$ is the Lagrange multiplier, and the adversarial perturbation is defined as

$$r_{adv} = \arg\max_{r; 0 < \|r\|_2} \frac{|f(x) - f(x + r)|}{\|r\|_2}. \qquad (18)$$

This formulation of WGAN results in a stable explicit Lipschitz penalty, overcoming the difficulties experienced when one tries to apply it to random sample pairs as shown in Petzka et al. (2018).

To evaluate the performance of WGAN-ALP, we trained one on CIFAR-10, consisting of $32 \times 32$ RGB images, using the residual architecture from Gulrajani et al. (2017), implemented in TensorFlow. Closely following Gulrajani et al. (2017), we used the Adam optimizer (Kingma and Ba, 2015) with parameters $\beta_1 = 0$, $\beta_2 = 0.9$ and an initial learning rate of $2 \times 10^{-4}$ decaying linearly to 0 over 100000 iterations, training the critic for 5 steps and the generator for 1 per iteration with minibatches of size 64 (doubled for the generator). We used (17) as a loss function to optimize the critic. $K = 1$ was an obvious choice, and we found $\lambda = 100$ to be optimal (the training diverged for $\lambda = 0.1$, and

was stable but performed worse for $\lambda = 10$ and $1000$). The hyperparameters of the approximation of $r_{adv}$ were set to $\xi = 10$, $P_\epsilon$ being the uniform distribution over $[0.1, 10]$, and $k = 1$ power iteration. Both batches from $P_r$ and $P_g$ were used for regularization.

We used Inception Score (Salimans et al., 2016) and FID (Heusel et al., 2017) as our evaluation metrics. The former correlates well with human judgment of image quality and is the most widely used among GAN models, and the latter has shown to capture model issues such as mode collapse, mode dropping and overfitting, while being a robust and efficient metric (Xu et al., 2018). We monitored the Inception Score and FID during training using 10000 samples every 1000 iteration, and evaluated them at the end of training using 50000 samples. We ran the training setting described above 10 times with different random seeds, and calculated the mean and standard deviation of the final Inception Scores and FIDs, while also recording the maximal Inception Score observed during training. We report these values for WGAN-ALP and other relevant GANs (Gulrajani et al., 2017; Petzka et al., 2018; Zhou et al., 2019a; Wei et al., 2018; Miyato et al., 2018; Adler and Lunz, 2018; Karras et al., 2018) in Table 1. We did not run experiments to evaluate competing models, but included the values reported in the corresponding papers (with the exception of the FID for WGAN-GP, which was taken from Zhou et al. (2019a)). They used different methods to arrive at the cited results, from which that of Adler and Lunz (2018) is the one closest to ours. We show some generated samples in Figure 1a.

Table 1: Inception Scores and FIDs on CIFAR-10

| Method | Inception Score | | FID |
| --- | --- | --- | --- |
| | Average | Best | |
| WGAN-GP | $7.86 \pm .07$ | | $18.86 \pm .13$ |
| WGAN-LP | $8.02 \pm .07$ | | |
| LGAN | $8.03 \pm .03$ | | $15.64 \pm .07$ |
| CT-GAN | $8.12 \pm .12$ | | |
| SN-GAN | $8.22 \pm .05$ | | $21.70 \pm .21$ |
| BWGAN | $8.31 \pm .07$ | | $16.43$ |
| Progressive GAN | $8.56 \pm .06$ | $8.80$ | |
| **WGAN-ALP (ours)** | $\mathbf{8.34 \pm .06}$ | **8.59** | $\mathbf{12.96 \pm .35}$ |

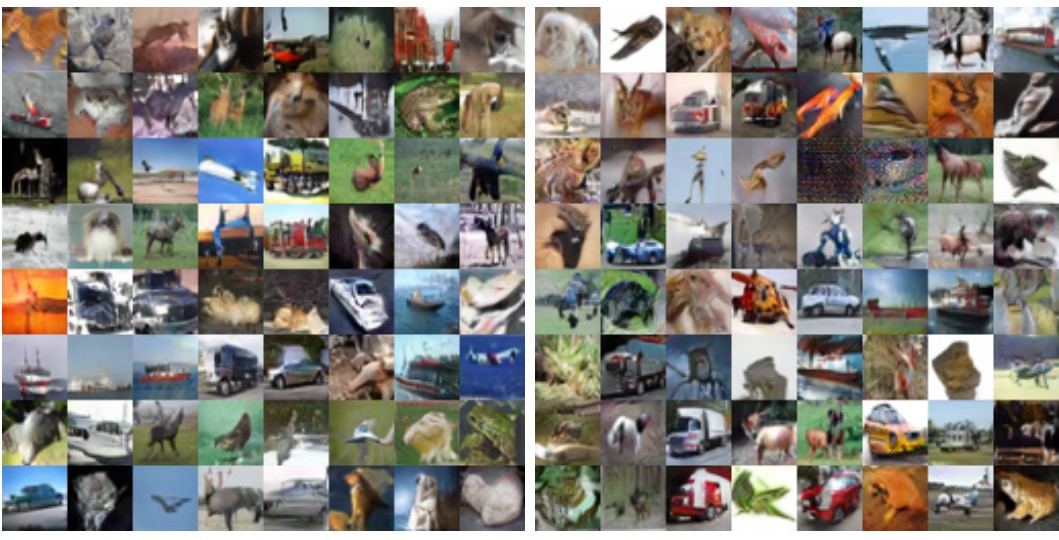

(a) No BN in critic          (b) BN in critic

Figure 1: Generated CIFAR-10 samples

We also trained WGAN-LP in our implementation. During training, the best observed Inception Score and FID were 8.13 and 18.49, while at the end of training the best final Inception Score and

FID were 8.01 and 15.42. To see that ALR indeed restricts the Lipschitz constant of the critic, we monitored the gradient norms during training, which converged to $\approx 5$ with $\lambda = 100$. This was also the case using LP with $\lambda = 0.1$, but the number of Lipschitz constraint violations found by the algorithm were much higher in this case than with ALR.

Our toy example in Appendix A.4 showed that when the regularized network contains BN layers, ALR seems to work better than competing methods. In order to see if this still applies in more complex settings, we have trained a variant of WGAN in which the critic contains BN layers (WGAN-BN). Gulrajani et al. (2017) did not use BN in the critic as they argued that GP is not valid in that setting, and indeed when we trained WGAN-BN with GP, the best Inception Score observed during training was only 6.29. When we applied ALP to WGAN-BN, the results were nearly on par with the original setting without BN, producing an even better maximal Inception Score of $8.71$. We leave the question of how BN affects Lipschitz continuity for future work. Generated samples are shown in Figure 1b.

Gulrajani et al. (2017) made the distinction between one-sided and two-sided penalties, represented by (9) and (7). The latter is based on the fact that in WGAN, the optimal critic has unit gradient norm on lines connecting points from the optimal coupling $\pi^*$. Petzka et al. (2018) showed that since $\pi^*$ is not known in practice, one should use the one-sided penalty, while Gemici et al. (2018) proposed a method to approximate $\pi^*$ with an auto-encoding scheme. In the limit $\|r\|_2 \to 0$ the expression inside the $\arg\max$ operator in (18) is equivalent to the directional derivative of $f$ along $r$, and the vector $r_{adv}$ corresponding to the maximum value of the directional derivative at $x$ is equivalent to $\nabla_x f(x)$. Since the critic $f$ corresponds to the potential function in the dual formulation of the optimal transport problem, at optimality its gradient at $x$ points towards its coupling $y$, where $(x, y) \sim \pi^*$. From this perspective, sampling pairs $(x, x + r_{adv})$ using (18) can be seen as an approximation of the optimal coupling $\pi^*$. To test how reasonable this approximation is, we have trained a WGAN variant with the two-sided explicit penalty formulated as

$$\mathbb{E}_{z \sim P_Z} f(g(z)) - \mathbb{E}_{x \sim P_r} f(x) + \lambda \mathbb{E}_{x \sim P_{r,g}} \left( \frac{|f(x) - f(x + r_{adv})|}{\|r_{adv}\|_2} - 1 \right)^2, \qquad (19)$$

which performed similarly to the one-sided case with $\lambda = 10$, but was less stable for other values of $\lambda$. The findings of Petzka et al. (2018) were similar for the case of the implicit penalty. Improving the approximation scheme of $r_{adv}$ might render the formulation using the two-sided penalty (19) preferable in the future.

To show that ALR works in a high-dimensional setting as well, we trained a Progressive GAN on the CelebA-HQ dataset (Karras et al., 2018), consisting of $1024 \times 1024$ RGB images. We took the official TensorFlow implementation and replaced the loss function of the critic, which originally used GP, with a version of ALP. Using (17) as the training objective was stable until the last stage of progressive growing, but to make it work on the highest resolution, we had to replace it with

$$\mathbb{E}_{z \sim P_Z} f(g(z)) - \mathbb{E}_{x \sim P_r} f(x)$$
$$+ \lambda \mathbb{E}_{x \sim P_{r,g}} \left( \left( \frac{|f(x) - f(x + r_{adv})|}{\|r_{adv}\|_2} - 1 \right)_+^2 + \left( \frac{|f(x) - f(x + r_{adv})|}{\|r_{adv}\|_2} - 1 \right)_+ \right), \quad (20)$$

meaning that we used the sum of the absolute and squared values of the Lipschitz constraint violation as the penalty. The optimal hyperparameters were $\lambda = 0.1$, $P_\epsilon$ being the uniform distribution over $[0.1, 100]$, $\xi = 10$ and $k = 1$ step of power iteration. The best FID seen during training with the original GP version was $8.69$, while for the modified ALP version it was $14.65$. The example shows that while ALP did not beat GP in this case (possibly because the implementation was fine-tuned using GP), it does work in the high-dimensional setting as well. For samples generated by the best performing ALR and GP variants see Appendix A.5.

## 5  CONCLUSIONS

Inspired by VAT, we proposed ALR and shown that it is an efficient and powerful method for learning Lipschitz constrained mappings implemented by neural networks. Resulting in competitive performance when applied to the training of WGANs, ALR is a generally applicable regularization method. It draws an important parallel between Lipschitz regularization and adversarial training, which we believe can prove to be a fruitful line of future research.

ACKNOWLEDGEMENTS

The author would like to thank Michael Herman from Bosch Center for Artificial Intelligence (BCAI) for the fruitful discussions, and the Advanced Engineering team in Budapest, especially Géza Velkey.

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

# A  APPENDIX

## A.1  SEMI-SUPERVISED LEARNING

Since VAT is a semi-supervised learning method, it is important to see how ALR fares in that regime. To show this, we have replicated one of the experiments from Miyato et al. (2019). We trained the ConvLarge architecture to classify images from CIFAR-10 with the same setting as described in Miyato et al. (2019), except that we did not decay the learning rate, but kept it fixed at $3 \times 10^{-4}$. We split the 50000 training examples into 4000 samples for the classification loss, 45000 samples for regularization and 1000 for validation, with equally distributed classes. Test performance was evaluated on the 10000 test examples. We have found that unlike in the unsupervised setting, here it was important to assume $f(x)$ fixed when minimizing the regularization loss, and also to complement the smoothing effect with entropy minimization (Grandvalet and Bengio, 2004). The baseline VAT method was ALR specialized with $K = 0$, $d_X$ being the Euclidean metric, $d_Y$ being the KL divergence, fixed $\epsilon = 8$ and $\lambda = 1$. This setting achieved maximal validation performance of 84.2% and test performance 82.46%. After some experimentation, the best performing choice was $K = 0$, $d_X$ being the $l_2$ metric, $d_Y$ the mean squared difference over the logit space (which parametrize the categorical output distribution over which the KL divergence is computed in the case of VAT), $P_\epsilon$ being the uniform distribution over $[1, 10]$ and $\lambda = 1$. This way the maximal validation performance was 85.3% and test performance 83.54%. Although this $\approx 1\%$ is improvement is not very significant, it shows that ALR can be a competitive choice as a semi-supervised learning method as well.

## A.2  VIRTUAL ADVERSARIAL TRAINING AS LIPSCHITZ REGULARIZATION

VAT was defined by considering neural networks implementing conditional distributions $p(y|x)$, where the distribution over discrete labels $y$ was conditioned on the input image $x$ Miyato et al. (2019). To see why LDS (10), the regularization term of VAT, can be seen as special kind of Lipschitz continuity, we will use a different perspective. Consider a mapping $f : X \to Y$ with domain $X$ and codomain $Y$, where $X$ is the space of images and $Y$ is the probability simplex (the space of distributions over the finite set of labels).

Since a divergence is in general a premetric (prametric, quasi-distance) on the space of probability measures (Deza and Deza, 2009), and Lipschitz continuity is defined for mappings between metric spaces, let us restrict the divergence $D$ from the VAT formulation to be a metric $d_Y$. Miyato et al. (2019) used KLD in their experiments, which is not a metric, but one can use e.g. the square root of JSD or the Hellinger distance, which are metrics. Let us metrize the space of images $X$ with $d_X$ being the Euclidean metric. From this perspective, the network $f$ is a mapping from the metric space $(X, d_X)$ to the metric space $(Y, d_Y)$. Let us also assume that we aim to learn a mapping $f$ with the smallest possible $\|f\|_L$ by setting $K$ to 0.

To enforce the condition $x + r \in X$ in (14), we bound the Euclidean norm of $r$ from above by some predefined $\epsilon > 0$. If we make the additional assumption that the supremum is always achieved with an $r$ of maximal norm $\epsilon$, the denominator in (14) will be constant, hence the formulas with and without it will be equivalent up to a scaling factor. With these simplifications, (14) and (15) reduce to

$$r_{adv}^{VAT} = \underset{0 \leq \|r\|_2 \leq \epsilon}{\arg\max} \, d_Y(f(x), f(x + r)) \tag{21}$$

and

$$L_{ALP}^{VAT} = d_Y(f(x), f(x + r_{adv})), \tag{22}$$

which are equivalent to (11) and (10), respectively. Let us consider the question of keeping $f(x)$ fixed when minimizing (22) an implementation detail. With this discrepancy aside, we have recovered VAT as a special case of Lipschitz regularization.

## A.3  DERIVATION OF THE APPROXIMATION OF $r_{adv}$

We assume that $f$ and $d_Y$ are both twice differentiable with respect to their arguments almost everywhere, the latter specifically at $x = y$. Note that one can easily find a $d_Y$ for which the last assumption does not hold, for example the $l1$ distance. If $d_Y$ is translation invariant, meaning that

$d_Y(x, y) = d_Y(x + u, y + u)$ for each $u \in Y$, then its subderivatives at $x = y$ will be independent of $x$, hence the method described below will still work. Otherwise, one can resort to using a proxy metric in place of $d_Y$ for the approximation, for example the $l2$ distance.

We denote $d_Y(f(x), f(x + r))$ by $d(r, x)$ for simplicity. Because $d(r, x) \geq 0$ and $d(0, x) = 0$, it is easy to see that

$$\nabla_r d(r, x)\big|_{r=0} = 0, \tag{23}$$

so that the second-order Taylor approximation of $d(r, x)$ is $d(r, x) \approx \frac{1}{2} r^T H(x) r$, where $H(x) = \nabla \nabla_r d(r, x)\big|_{r=0}$ is the Hessian matrix. The eigenvector $u$ of $H(x)$ corresponding to its eigenvalue with the greatest absolute value is the direction of greatest curvature, which is approximately the adversarial direction that we are looking for. The power iteration (Householder, 1964) defined by

$$r_{i+1} := \frac{H(x) r_i}{\|H(x) r_i\|_2}, \tag{24}$$

where $r_0$ is a randomly sampled unit vector, converges to $u$ if $u$ and $r_0$ are not perpendicular. Calculating $H(x)$ is computationally heavy, which is why $H(x) r_i$ is approximated using the finite differences method as

$$H(x) r_i \approx \frac{\nabla_r d(r, x)\big|_{r=\xi r_i} - \nabla_r d(r, x)\big|_{r=0}}{\xi} = \frac{\nabla_r d(r, x)\big|_{r=\xi r_i}}{\xi} \tag{25}$$

where the equality follows from (23). The hyperparameter $\xi \neq 0$ is introduced here. In summary, the adversarial direction is approximated by the iterative scheme

$$r_{i+1} := \frac{\nabla_r d(r, x)\big|_{r=\xi r_i}}{\left\| \nabla_r d(r, x)\big|_{r=\xi r_i} \right\|_2}, \tag{26}$$

of which one iteration is found to be sufficient and necessary in practice.

## A.4 Toy example

To showcase the differences between weight normalization methods, implicit penalty methods and explicit penalty methods, represented by SN, LP and ALR, respectively, we devised the following toy example. Suppose that we want to approximate the following real-valued mapping on the 2-dimensional interval $[-4, 4]^2$:

$$f(x, y) = \begin{cases} 0 & \text{if } 1 \leq \sqrt{x^2 + y^2} \leq 2, \\ 1 & \text{otherwise} \end{cases} \tag{27}$$

for $-4 \leq x, y \leq 4$. In addition, we want the approximation to be 1-Lipschitz. It is easy to see that the optimal approximation with respect to the mean squared error is

$$\hat{f}_{opt}(x, y) = \begin{cases} 1 & \text{if } \sqrt{x^2 + y^2} \leq 0.5, \\ 1.5 - \sqrt{x^2 + y^2} & \text{if } 0.5 < \sqrt{x^2 + y^2} \leq 1.5, \\ \sqrt{x^2 + y^2} - 1.5 & \text{if } 1.5 < \sqrt{x^2 + y^2} \leq 2.5, \\ 1 & \text{otherwise.} \end{cases} \tag{28}$$

This example has connections to WGAN, as the optimal critic is 1-Lipschitz, and its approximation will provide the learning signal to the generator in the form of gradients. Therefore, it is important to closely approximate the gradient of the optimal critic, which is achieved indirectly by Lipschitz regularization. In this example, we will see how closely the different Lipschitz regularization methods can match the gradient of the optimal approximation $\hat{f}_{opt}$.

We implemented the example in PyTorch. For the approximation $\hat{f}$, we use an MLP with 3 hidden layers containing 20, 40 and 20 neurons, respectively, with ReLU activations after the hidden layers, and a variant which also has batch normalization (BN) before the activations, since it has been found that BN hurts adversarial robustness (Galloway et al., 2019), and hence it should also hurt Lipschitz continuity. We trained the networks for $2^{14}$ iterations, with batches consisting of an input, a

corresponding output, and an additional input for regularization. The inputs are drawn uniformly at random from $[-4, 4]^2$ and the output is defined by (27). The minibatch size was 64 for input-output pairs, and 1024 for regularization inputs. We used heatmaps to visualize the gradient norm surfaces of the optimal and learned mappings, with the color gradient going from black at 0 to white at 1, see Figure 2. This example is not intended to rank the competing Lipschitz regularization methods, as it always depends on the particular application which one is the best suited, but to show that they are fundamentally different and competent in their own way.

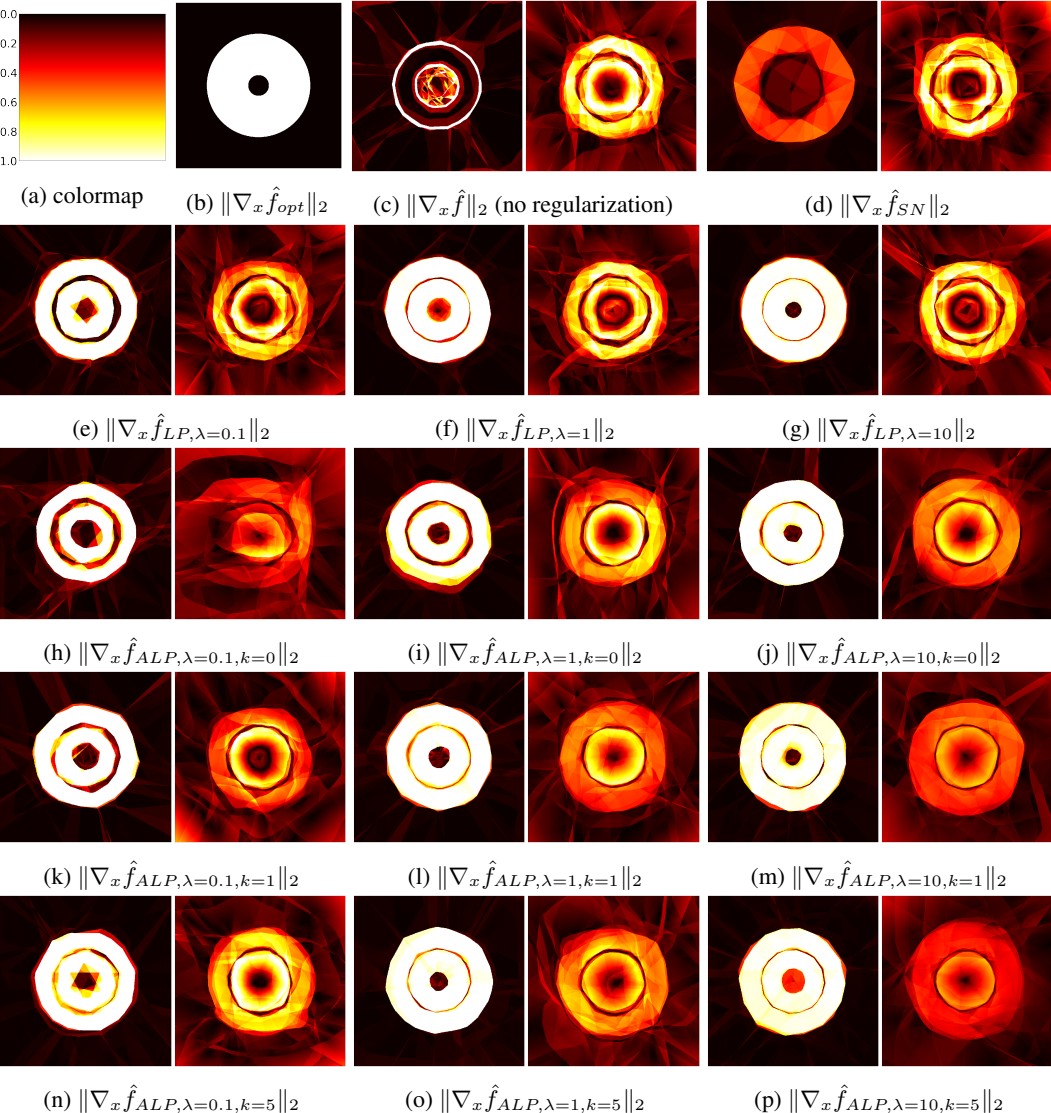

(a) colormap    (b) $\|\nabla_x \hat{f}_{opt}\|_2$    (c) $\|\nabla_x \hat{f}\|_2$ (no regularization)    (d) $\|\nabla_x \hat{f}_{SN}\|_2$

(e) $\|\nabla_x \hat{f}_{LP,\lambda=0.1}\|_2$    (f) $\|\nabla_x \hat{f}_{LP,\lambda=1}\|_2$    (g) $\|\nabla_x \hat{f}_{LP,\lambda=10}\|_2$

(h) $\|\nabla_x \hat{f}_{ALP,\lambda=0.1,k=0}\|_2$    (i) $\|\nabla_x \hat{f}_{ALP,\lambda=1,k=0}\|_2$    (j) $\|\nabla_x \hat{f}_{ALP,\lambda=10,k=0}\|_2$

(k) $\|\nabla_x \hat{f}_{ALP,\lambda=0.1,k=1}\|_2$    (l) $\|\nabla_x \hat{f}_{ALP,\lambda=1,k=1}\|_2$    (m) $\|\nabla_x \hat{f}_{ALP,\lambda=10,k=1}\|_2$

(n) $\|\nabla_x \hat{f}_{ALP,\lambda=0.1,k=5}\|_2$    (o) $\|\nabla_x \hat{f}_{ALP,\lambda=1,k=5}\|_2$    (p) $\|\nabla_x \hat{f}_{ALP,\lambda=10,k=5}\|_2$

Figure 2: Gradient norm surfaces of optimal and learned approximations of $f$

Without any kind of regularization, the network learned to approximate the target function very well, but its gradients look nothing like that of $\hat{f}_{opt}$, although somehow it is a better match with BN.

When we apply SN to the MLP layers, the result without BN will be a very smooth mapping with maximum gradient norm far below 1. SN is not compatible with BN, the result being only slightly better than the unregularized case. A detail not visible here is that because SN considers weight matrices as linear maps from $\mathbb{R}^n$ to $\mathbb{R}^m$ and normalizes them layer-wise, it regularizes globally instead of around actual data samples. In this case, on the whole of $\mathbb{R}^2$ instead of just $[-4, 4]^2$. For WGANs trained on CIFAR-10, the input space consists of $32 \times 32$ RGB images with pixel values in $[-1, 1]$, but the trained mapping is regularized on $\mathbb{R}^{32 \times 32 \times 3}$ instead of just $[-1, 1]^{32 \times 32 \times 3}$ (which

contains the supports of the real and fake distributions). This can hurt performance if the optimal mapping implemented by a particular network architecture is K-Lipschitz inside these supports, but not in some other parts of $\mathbb{R}^{32 \times 32 \times 3}$.

When the network is regularized using LP (9), the regularization strength can be controlled by tuning the value of $\lambda$. We trained with $\lambda = 0.1, 1$ and $10$. Without BN, the highest of these values seems to work the best. With BN, the resulting mapping is visibly highly irregular.

With ALR, in addition to $\lambda$, we have additional control over the regularization by the hyperparameters of the approximation scheme of $r_{adv}$. After some experimentation, we have found the best $P_\epsilon$ for this case was the uniform distribution over $[10^{-6}, 10^{-5}]$. We trained with $\lambda = 0.1, 1$ and $10$, and $k = 0, 1$ and $5$ power iterations. Arguably, both with and without BN the $\lambda = 1$ and $k = 5$ case seems like the best choice. Without BN, the results are quite similar to the LP case, but when BN is introduced, the resulting mappings are much smoother than the ones obtained with LP.

A.5 IMAGES GENERATED BY PROGRESSIVE GAN TRAINED ON CELEBA-HQ

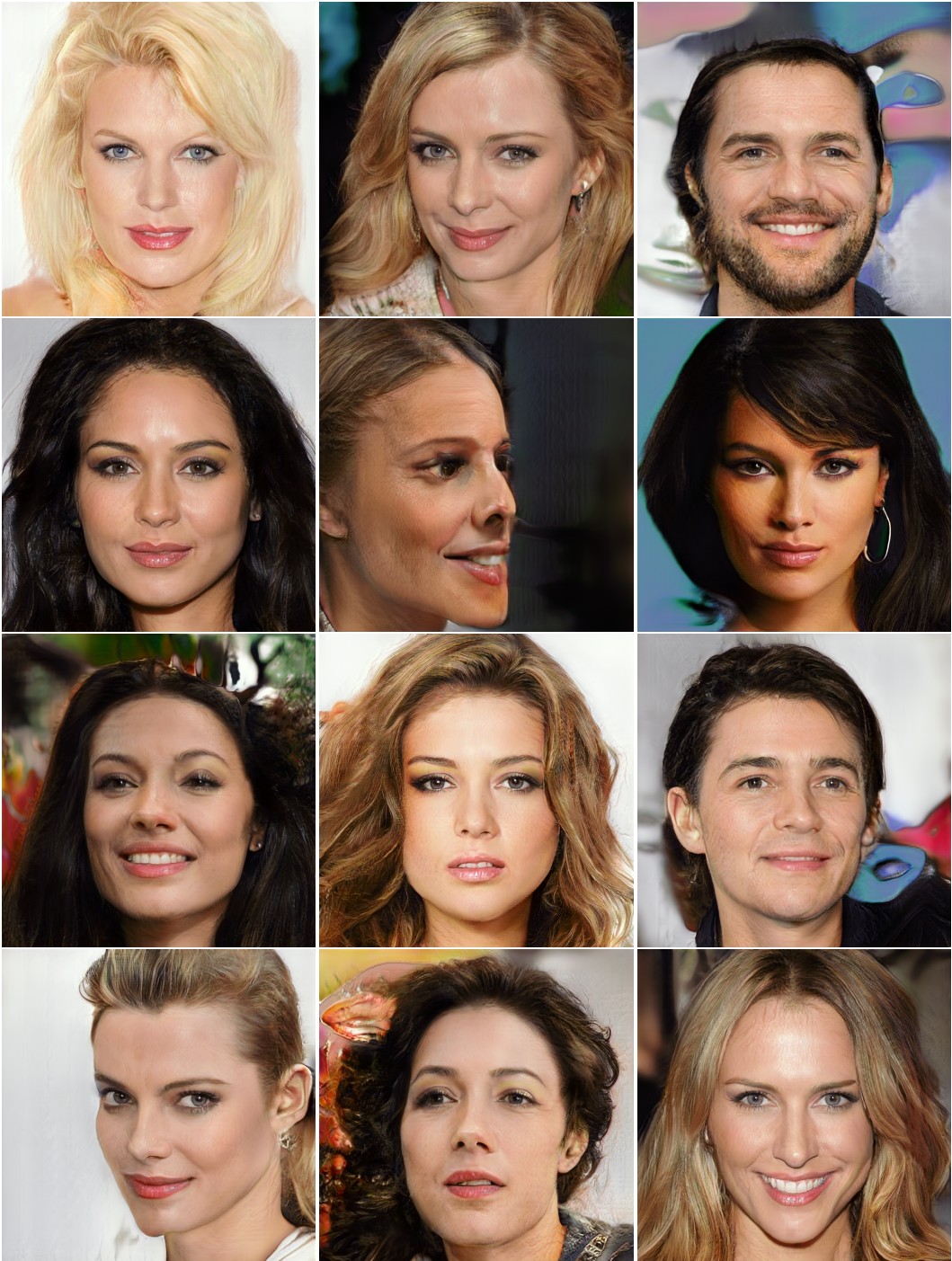

Figure 3: Images generated using Progressive GAN trained with ALR

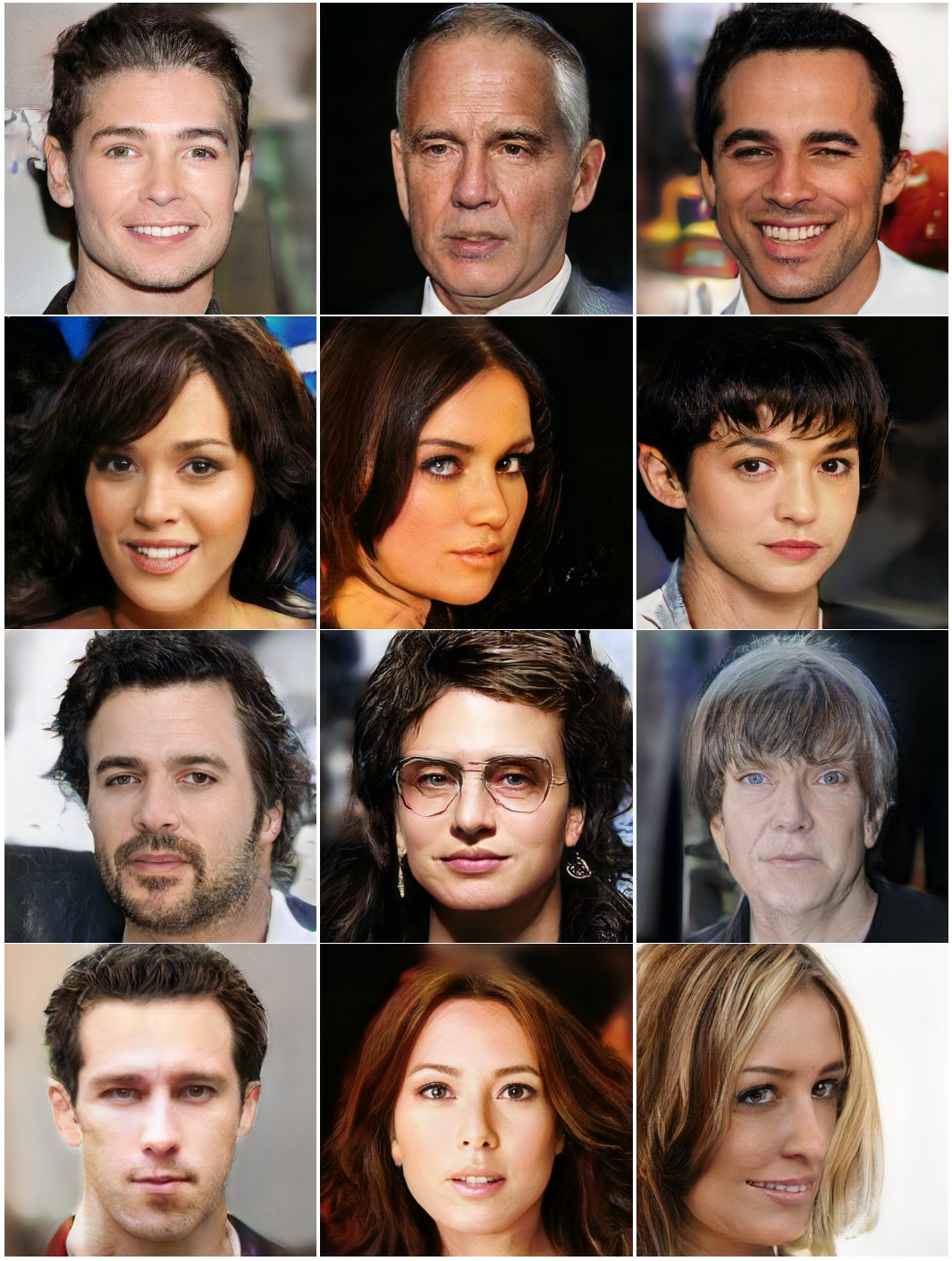

Figure 4: Images generated using Progressive GAN trained with GP

