# OpenReview forum: "Adversarial Lipschitz Regularization"
_ICLR.cc/2020/Conference — Accept (Poster)_

### Official Review · AnonReviewer1 · 2019-10-25
**Official Blind Review #1**

**Rating:** 6

**Review:**

It is an interesting idea about how to enforce the Lipsthitz constrain in WGAN by using virtual adversarial training. The connection between virtual adversarial and this paper method - ALR is quite simple and clear. In the experiments, the FID score in the table is not complete which can not clearly compare the ability of the Lipschitz regularization to other regularization methods. The paper addresses that the approximation of r_{adv} will affect the performance of ALR. How to balance the quality and computation complexity is quite important. This paper did not provide the reason about why this method can not work better than GP method in high-dimensional setting.
In general, this paper provides an interesting direction for regularization.

Pros:
1. This paper derived as a generalization of VAT (Virtual Adversarial Training) which provided the new way to think of the regularization.
2. ALR (Adversarial Lipchitz Regularization) is an new method for learning Lipschitz constrained rather than weight clipping or gradient penalty.
3. This method provides the connection between Lipschitz regularization and adversarial training.

Cons:
1. The comparison of the experiments was not complete. Some of the Inception Scores and FID were blank in the table.
2. The results of adding BN were not clearly explained. These included LP and ALR method. Might have some inference about the effect of BN in regularization term.
3. In high-dimensional setting, the authors did not clearly describe the weakness of ALR method.

**Experience Assessment:**

I have published one or two papers in this area.

**Review Assessment: Checking Correctness Of Derivations And Theory:**

I assessed the sensibility of the derivations and theory.

**Review Assessment: Checking Correctness Of Experiments:**

I did not assess the experiments.

**Review Assessment: Thoroughness In Paper Reading:**

I made a quick assessment of this paper.

---

> ### Author Response · Authors · 2019-11-12
> **Clarification regarding evaluation**
>
> Dear Reviewer,
>
> First of all, thank you for taking the time and reviewing our paper.
>
> While we are working on a revision, we would like to clarify the issues regarding the evaluation. Table 1 is not complete because the cited papers did not always reported best Inception Score or Frechet Inception Distance. We did train WGAN-LP in our implementation with \lambda = 0.1 and 10 (both 2 times). The best final IS is 8.009, while the best final FID is 15.42. During training, the best observed IS was 8.127, while the best FID was 18.49. We'd like to include these in the revision as well.

---

> ### Author Response · Authors · 2019-11-14
> **Revision**
>
> Dear Reviewer,
>
> We have uploaded a revision which incorporates the feedback you have given. Specifically,
>
> 1. We have added a discussion about Table 1. Only the results of WGAN-ALP are ours, the others were taken from the cited papers.
>
> 2. We have explored the usage of BN a bit more, showing that adding BN to the critic in WGAN works with ALR, while it does not work very well with GP.
>
> 3. We believe there is no apparent weakness of ALR in the high dimensional setting, and hypothesize that ALR performing worse than GP is caused by the fact that the official Progressive GAN implementation was fine-tuned using GP, and we did not change anything other than the regularization method. We shortly discuss this in the paper as well.
>
> Thank you again for the review, it has been very helpful in improving our paper.

---

### Official Review · AnonReviewer4 · 2019-11-02
**Official Blind Review #4**

**Rating:** 6

**Review:**

Summary: Virtual Adversarial Training (Miyato et al., 2017) can be viewed as a form of Lipschitz regularization. Inspired by this, the paper proposes a Lipschitz regularization technique that tries to ensure that the function being regularized doesn’t change a lot in virtual adversarial directions. This method is shown to be effective in training Wasserstein GANs.

Motivation and placement in literature: Being able to effectively enforce the Lipschitz constraint on neural networks has wide ranging applications. Even though the paper predominantly considers the WGAN setting, the topic at hand is within the scope of NeurIPS and will of interest to the machine learning community at large.

Claimed Contributions and their significance:
1. Practical method with good performance: The proposed method can be used to train WGANs with high (subjective) sample quality. Although better, quantitative evaluation methods are needed to make stronger claims about the efficacy of this approach for GAN training in general (see below), the method described here will likely be useful for practitioners and GAN community. I’m also convinced that this method has the potential to work for higher dimensions.
2. VAT as Lipschitz regularization: There is a relatively straightforward connection between the Lipschitz constraint and adversarial robustness - both imply that small changes in the inputs should lead to small changes in the outputs, in their respective space. There are also a number of papers that make strong connections between adversarial training and Lipschitz regularization (Parseval Networks (Cisse et. al, 2017) for example). Therefore, it is perhaps not too surprising that the LDS term from Miyato et. al. can be rephrased as a Lipschitz regularization term by picking suitable input-output (pre-)metrics (in Section 3). I currently don’t see this as a major contribution of this paper, although I’m open to changing my mind if this involves a subtlety that I’m missing.

Related Work:
Khrulkov et al (2017) looks like a related work - especially related to how the way the adversarial perturbation is computed and backpropagation is performed. Also Gemici et. al. also discuss the limitations of the original gradient penalty paper (for Section 2.2)

Questions and Points of Improvement
1. Better evaluation of GANs: Could you further convince us that this method alleviates common pitfalls of GAN training, such as mode collapse? There are a number of papers that give quantitative metrics for this purpose (such as Xu et. al. 2018). Since the quality of the WGANs presented is one of the biggest strengths of this paper, further evidence in this direction will make the paper stronger.

2. Different tasks:
The method described looks flexible enough to be applied on domains other than Wasserstein distance estimation. Did you try other tasks where a Lipschitz penalty might help, such as adversarial robustness? The semi-supervised setting mentioned in the appendix look promising yet perhaps under-explored.

3. Resultant Lipschitz constant:
Since this paper is about enforcing the Lipschitz constraint through regularization, more experiments on how well the Lipschitz constraint is enforced in practise would be helpful. For example, how much do your WGAN critics violate the 1-Lipschitz constraint? Once this is quantified, how does ALR compare to other Lipschitz regularization techniques? The function approximation task in Section 4.2 seems simple enough that you can probably compute gradient norms on a 2D grid and draw a histogram. How would the histograms look if you did this, for different methods?

4. Sample efficiency:
Section 4.2 claims that using the explicit Lipschitz penalty is inefficient because violations of the Lipschitz constraint on samples from P_real, P_generated or P_interpolated likely be non-maximal. Could you make a theoretical or empirical case that the additional time spent for finding adversarial directions is actually worth it? If you have a way of quantifying how well the Lipschitz constraint is satisfied (as described above), then doing this empirically should be possible.

5. Problematic baseline for spectral normalization:
The way spectral normalization (SN) was used/described in Section 4.1 seem to have some issues. First of all, batch normalization is incompatible with methods that achieve Lipschitz constraint via. architectural constraints, such as spectral normalization. Also, this statement looks problematic: “It can be seen that SN has a very strong regularization effect, which is because SN works by approximating the spectral norm of each layer and then normalizing the layers by dividing their weight matrices by the corresponding spectral norms, thereby resulting in overregularization if the approximation is greater than the actual spectral norm.“ In most practical cases, power iteration used in spectral normalization can get a very close approximation of the spectral norm of the weight matrices with a reasonable number (<20 is a conservative guess) of iterations. The over-regularization effect, however, does exist and is more connected to the loss of gradient norm as described in Anil et. al. than bad approximations to the spectral norm of weight matrices.

Writing: The paper is well-written and easy to understand.

Decision: Weak Accept.

Other, lesser important points of improvement:
1. The argmax expression in (18) looks problematic - r doesn’t seem bounded, hence can be chosen arbitrarily large.
2. Equation (25) describes the optimal approximation. According to which metric is this optimal?
3. Use \leq for “less than or equal to” in 25.
4. Consider adding a colormap to Figure 1.

________
Post-rebuttal edit: The revisions made to the paper address some of the points of improvement listed above. I maintain my initial assessment of weak accept (leaning more towards accept), as I believe the methods discussed in this paper will be of interest to the research community.


**Experience Assessment:**

I have published one or two papers in this area.

**Review Assessment: Checking Correctness Of Derivations And Theory:**

I assessed the sensibility of the derivations and theory.

**Review Assessment: Checking Correctness Of Experiments:**

I assessed the sensibility of the experiments.

**Review Assessment: Thoroughness In Paper Reading:**

I read the paper thoroughly.

---

> ### Author Response · Authors · 2019-11-12
> **Question regarding citations and clarification regarding evaluation**
>
> Dear Reviewer,
>
> First of all, thank you for taking the time and reviewing our paper.
>
> While we are working on a revision, we would like to make sure that we got the correct papers for the following citations:
> Khrulkov et al (2017) - "Art of singular vectors and universal adversarial perturbations"
> Gemici et. al. - "Primal-Dual Wasserstein GAN"
> Xu et. al. 2018 - "An empirical study on evaluation metrics of generative adversarial networks"
> Just to make sure, are these the papers you were referring to?
>
> Regarding evaluation, we see that it would be an important improvement to evaluate the WGAN models with metrics other than IS and FID, such as MMD for example. Unfortunately this is currently a stretch goal for us as it is not sure that we can do this until the rebuttal deadline. But, in the paper "An empirical study on evaluation metrics of generative adversarial networks" by Xu et al. (2018), it is shown that FID successfully captures mode collapse (see Figure 3), as well as other model issues. They conclude by saying "Fréchet Inception Distance performs well in terms of discriminability, robustness and efficiency. It serves as a good metric for GANs, despite only modeling the first two moments of the distributions in feature space."

---

> > ### Comment · AnonReviewer4 · 2019-11-12
> > **Responding to author comments**
> >
> > Thank you for your response.
> >
> > Yes, those are the references I pointed at in my review.
> >
> > While I believe a more diverse set of evaluation metrics, combined with some discussion on the type of qualities they evaluate would improve the paper, I understand that this is difficult to complete before the rebuttal deadline and will take this into consideration when forming my final thoughts.

---

> ### Author Response · Authors · 2019-11-14
> **Revision**
>
> Dear Reviewer,
>
> We have uploaded a revision which incorporates the feedback you have given. Specifically,
>
> Claimed contributions and their significance:
> We do not consider "VAT as Lipschitz Regularization" as a contribution, and even moved the section describing it to the Appendix as it does not contribute much to the paper.
>
> Related work:
> We cited both Khrulkov et al. and Gemici et al. which are relevant work.
>
> Questions and points of improvement:
> 1. We have described the properties of Inception Score and FID. Unfortunately we did not have the time to implement other metrics.
>
> 2. We sadly did not have the time to explore the important semi-supervised direction further. We see Lipschitz regularization of neural networks other than WGAN an important research area, and we hope ALR contributes to it.
>
> 3. We did compute the gradient norms on a 2D grid and visualized them as heatmaps. We also described in Section 4 "WGAN-ALP" that by monitoring the gradient norms of the critic it is visible that ALR indeed effectively restricts the Lipschitz constant.
>
> 4. Defining ALR with k=0 steps of power iteration effectively results in random perturbations, which were evaluated by Petzka et al. They have found that it makes one unable to train WGANs on CIFAR-10 using the explicit penalty. Doing at least 1 step of power iteration produces better perturbations and in turn makes the training work.
>
> 5. We have revised the discussion of SN in the toy example.
>
> All minor comments have been incorporated as well.
>
> Thank you again for the review, it has been very helpful in improving our paper.

---

### Official Review · AnonReviewer5 · 2019-11-04
**Official Blind Review #5**

**Rating:** 6

**Review:**

Final Edit:

I have reviewer the final version of the paper and have decided to increase my score to a weak accept. I maintain some concerns around the empirical evaluation in the paper (collating results from multiple sources with different experimental procedures). But my major concerns have been addressed by the authors in their response and changes to the paper.

----------------------------------
Post rebuttal edit:

Following updates to the paper manuscript, which address my concerns around the correctness of the empirical results, I have updated my review score from 1 to 3.
----------------------------------

Summary:

This paper draws on connections between virtual adversarial training (VAT) and Lipschitz regularization to utilize VAT techniques in the training of WGAN architectures. While this work may be touching on something quite interesting, I felt that the theoretical exposition of the ideas was lacking. The empirical evidence seemed promising in some direction though lacking in others.

I was requested as an emergency reviewer for this paper.


Overview:

This paper is 9 pages length in total. Unfortunately, I felt that the use of an additional page was unwarranted and that the paper contained unnecessary content.

Due to a concern over correctness of some empirical results and issues with the presented derivations I have opted to reject this paper. I hope that these issues can be addressed by the authors in which case I will reassess my score.

1) Under equation 5, "with substantially more stable training behaviour and improved sample quality". A citation should be included for this claim. In fact, recent advances in GAN methods have not required the Wasserstein distance objective [1].

2) I found some issues with Section 2.2.

a) First a comment on related work. There is older work studying the generalization properties of Lipschitz neural networks which is not mentioned in this section. For example, [2]. You also write that learning under Lipschitz constraints became prevalent with the introduction of WGAN. While this is probably true, I think it is fair to point out that many older papers also utilized similar bounds in the vein of improving generalization. For example [3], which also aimed to limit the gradient norm of deep neural networks.

b) I felt that this subsection was a little bloated and the content did not fit entirely under the heading. A large chunk of this section is dedicated to discussing potential issues arising with the gradient penalty formulation of WGAN and alternative approaches such as the Banach WGAN. While these are useful additions they did not feel critical to this work and in my opinion did not deserve an extension beyond the 8 page recommendation.

3) I found the discussion in Section~3 a little difficult to follow. I will summarize my key concerns below.

a) The authors assume that generalizing Lipschitz continuity to a premetric space is trivial. While the more important results seems believable I am not convinced by the presentation of these results and would prefer to have seen this given more careful treatment. For example, premetrics need not obey symmetry or the triangle inequality (assuming this is the definition used by the authors --- including this would be valuable). It is written (paraphrasing) that a mapping $f$ is $K$-Lipschitz iff for any $x$ the supremum over $r$ is bounded by $K$. However, $r$ only appears on the right-hand side of a potentially asymmetric distance function. Moreover, many properties of Lipschitz continuous functions depend on the triangle inequality holding in the metric space and these would fail to hold here.

b) When connecting ALP to VAT some of the differences in the formulations are hand-waved away in unconvincing ways. Under the trivial metric, the Lipschitz constant is given by the maximum distance in the output space. With this observation, it seems trivial that the VAT formulation will perform a form of Lipschitz regularization. However, the Lipschitz constant does not take into account distance in the input space in a meaningful way and so I am unsure to what extend the connection is really meaningful. Further, the $r < \epsilon$ constraint in the VAT model is treated as an inconvenient implementation detail but I am not convinced this is sufficient. Indeed, this $\epsilon$ could be used to bound the input deviations and thus could be seen as affecting the Lipschitz constant under a more reasonable metric.

4) I felt that Section~4.1 presented an important discussion coupled with an interesting toy problem to highlight benefits and shortcomings of the proposed method. However, this section was moderately long and contributed only a little towards understanding the practical settings users of ALR would care about in practice. I did not gain much intuition into how ALR might generalize to high dimensional settings and was concerned by the fact that the distribution $P_\epsilon$ used was heavily hand-engineered and did not match up with the ones used in later experiments.

a) I did not understand the comments that constraining the Lipschitz constant globally may be undesirable in WGANs. The dual optimization problem requires a Lipschitz constraint be enforced over the support of the distributions and we should not care outside of this region in any case (except in cases where the generator might move the support to a currently under-regularized region of the critic domain in which case a global constraint may be advantageous).

5) I am not particularly up to date with evaluation of GAN models but to me the presented results in the main paper looked mostly reasonable. Some major concerns did remain to me which I would appreciate being addressed by the authors.

a) I have one question on the reported "Best" inception score for the WGAN-ALP and Progressive-GAN models. You stated that each model was trained 5 times and reported the mean, standard deviation and best results. However, the difference between the best and average scores alone would constitute a higher standard deviation: $\sqrt{(8.80 - 8.56)^2 / 4} = 0.12$. Please can you clarify how exactly each of these values was computed?

b) In the main paper the authors write that ALP is able to work in high dimensional settings (though is not competitive with state of the art). In the appendix however the authors point out that they must make significant modifications to the training objective by including a squared Lipschitz constraint violation term (violating further the comparison to VAT). I do not consider this a huge issue, but it should be discussed in the main paper.

c) Finally, the authors employed a range of different hyperparameter settings through their experiments but gave little guidance on how to choose these settings in practice or how sensitive their proposed method is to changes in these hyperparameters. I believe that this would be a highly-valuable addition to the paper and would help distinguish this method from other training stabilization proposals.


Minor comments:

- In paragraph 1, you write that WGAN requires critic to consist of only 1-Lipschitz functions. This is true of the Wasserstein distance estimation problem but the WGAN only requires the correct gradient direction (scaling of the critic is fine).
- In summary points in intro, "ALR" is used before acronym is defined.
- Equation (12) and (13) are twice normalized (||r_k||^2=1 by definition). Similar issue in (22) and (23).
- First para of Section 3, you write "on the space of labels". Do you mean on the probability simplex?


References:

[1] "Large scale GAN training for high fidelity natural image synthesis", Brock, Donahue, and Simonyan
[2] "The sample complexity of pattern classification with neural networks", Bartlett
[3] "Double backpropagation increasing generalization performance", Drucker and LeCun


**Experience Assessment:**

I have read many papers in this area.

**Review Assessment: Checking Correctness Of Derivations And Theory:**

I carefully checked the derivations and theory.

**Review Assessment: Checking Correctness Of Experiments:**

I carefully checked the experiments.

**Review Assessment: Thoroughness In Paper Reading:**

I read the paper at least twice and used my best judgement in assessing the paper.

---

> ### Author Response · Authors · 2019-11-12
> **Clarification regarding evaluation**
>
> Dear Reviewer,
>
> First of all, thank you for taking the time and reviewing our paper.
>
> While we are working on a revision, we would like to clarify the issues regarding the evaluation metrics. The method we used to compute the values was the following:
>
> During training for 100000 iterations, after every 1000 iteration we generated 10000 images to compute the Inception Score and the Frechet Inception Distance. After training, we generated 50000 images to compute the final IS and FID. We did this 5 times. The best IS reported was computed during 1 of the 5 trainings from 10000 samples. To get the average IS and FID, we calculated the mean and std of the 5 final IS and FID values. This means that the best reported value was not included in calculating the average and std values, because it was computed during training from 10000 samples, and not after training from 50000 samples.
>
> We checked the cited papers to see how they computed the reported results, and found the following:
>
> WGAN-GP (Gulrajani et al., 2017): the method used to compute the IS is not described, the FID reported in our paper is from (Zhou et al., 2019a), see below how it was calculated
>
> WGAN-LP (Petzka et al., 2018): "The maximal scores reached in 100000 training iterations with different regularization parameters are reported in Table 1." "Table 1: Inception Score on CIFAR-10. Reported are the maximal mean values reached during training. Means are computed over 10 image sets, variances given in parenthesis." We included the maximal value from that table.
>
> LGAN (Zhou et al., 2019a): "We use 200,000 iterations for better convergence and use 500k samples to evaluate IS and FID for preferable stability." Again we included the best values.
>
> CT-GAN (Wei et al., 2018): "For model selection, we use the first 50,000 samples to compute the inception scores (Salimans et al., 2016), then choose the best model, and finally report the “test” score on another 50,000 samples."
>
> SN-GAN (Miyato et al., 2018): "Following the procedure in Salimans et al. (2016); Warde-Farley & Bengio (2017), we calculated the score for randomly generated 5000 examples from each trained generator to evaluate its ability to generate natural images. We repeated each experiment 10 times and reported the average and the standard deviation of the inception scores." "We computed the Fréchet inception distance between the true distribution and the generated distribution empirically over 10000 and 5000 samples."
>
> BWGAN (Adler and Lunz, 2018): "For evaluation, we report Fréchet Inception Distance (FID)[8] and Inception scores, both computed from 50K samples."
>
> Progressive GAN (Karras et al., 2018): "We report our scores in two different ways: 1) the highest score observed during training runs (here ± refers to the standard deviation returned by the inception score calculator) and 2) the mean and standard deviation computed from the highest scores seen during training, starting from ten random initializations."
>
> The closest to our method is that of BWGAN. We have since completed another 5 trainings with the same setup to have 10 in total, and the new values are the following:
> IS: 8.3447 +- 0.059503025133181
> FID: 12.961 +- 0.34952682300504
>
> Another method, closer in spirit to that of Progressive GAN, is to hand-pick the 5 best values from each training and calculate the mean and std from those. This way, we get the following:
> IS: 8.41598 +- 0.067931580284872
> FID: 16.9014 +- 0.3634391833581
> While IS is better because of the hand-picking, FID is worse. This is because these values are based on FIDs that are computed from 10000 samples instead of 50000 samples. See Figure 6 in "An empirical study on evaluation metrics of generative adversarial networks" by Xu et al. (2018), there it is visible that FID is getting smaller as the sample size used to evaluate it gets bigger, which explains why the FIDs computed from bigger samples are better. Also see the documentation of the official TensorFlow implementation of FID (https://github.com/tensorflow/tensorflow/blob/r1.8/tensorflow/contrib/gan/python/eval/python/classifier_metrics_impl.py, lines 466-471): "Note that when computed using sample means and sample covariance matrices, Frechet distance is biased. It is more biased for small sample sizes. (e.g. even if the two distributions are the same, for a small sample size, the expected Frechet distance is large). It is important to use the same sample size to compute frechet classifier distance when comparing two generative models."
>
> Based on these, we believe the original method we used is relatively sensible, but we are open to suggestions regarding other methods.
>
> We also trained WGAN-LP in our implementation with \lambda = 0.1 and 10 (both 2 times). The best final IS is 8.009, while the best final FID is 15.42. During training, the best observed IS was 8.127, while the best FID was 18.49. We'd like to include these in the revision as well.

---

> ### Author Response · Authors · 2019-11-14
> **Revision**
>
> Dear Reviewer,
>
> We have uploaded a revision which incorporates the feedback you have given. Specifically,
>
> 1) We added a citation to (Arjovsky et al., 2017) to support the calim, and mentioned that "recent GAN variants do not always use this objective", citing (Brock et al., 2019).
>
> 2)a) We cited the older papers noting that they also connected low Lipschitz constants with good generalization.
>
> 2)b) We removed the paragraph containing Banach WGAN.
>
> 3)a) VAT was defined with an arbitrary divergence D, so we restricted our discussion to divergences that are also metrics.
>
> 3)b) We worked out another perspective of VAT as Lipschitz regularization where the metric is not the trivial one but the Euclidean distance. We moved a shorter version of this section to Appendix A.2 "Virtual Adversarial Training as Lipschitz regularization".
>
> 4) We replaced P_\epsilon with the uniform distribution over [10^{-6}, 10^{-5}], and moved the whole toy example to Appendix A.4 "Toy example". Section 3.3 "Comparison with other Lipschitz regularization techniques" now contains only the key takeaways from the toy example.
>
> 4)a) We clarified this discussion.
>
> 5)a) We have described how we arrived at the results, and also that results for the other models are from the cited articles and were computed differently than ours.
>
> 5)b) We have added a discussion of using the regularization term as it is or its square (or both) to Section 3.2 "Hyperparameters".
>
> 5)c) We have added guidance in choosing the right hyperparameter values in Section 3.2 "Hyperparameters".
>
> All minor comments have been incorporated as well.
>
> Thank you again for the review, it has been very helpful in improving our paper.

---

### Author Response · Authors · 2019-11-14
**Revision**

We have uploaded a revision incorporating the feedback given by the reviewers. Key changes are the following:

*** Section 3 "Virtual Adversarial Training as Lipschitz Regularization" has been removed. Part of it was incorporated into Section 4 (now Section 3) "Adversarial Lipschitz Regularization", specifically the part that takes a mapping f between metric spaces and arrives at the notion of adversarial perturbation wrt. the Lipschitz continuity and the ALP loss term. The rest has been reworked to use the more sensible Euclidean metric as d_X instead of the trivial 0-1 metric, and for d_Y we now only consider divergences that are metrics as well, to avoid the difficulties arising when one tries to define Lipschitzness with premetrics. This part has been moved to Appendix A.2 "Virtual Adversarial Training as Lipschitz Regularization".

*** Section 3.2 "Hyperparameters" has been added which describes the hyperparameters of ALR and gives some guidance towards tuning them.

*** Section 3.3 "Comparison with other Lipschitz regularization techniques" now contains only the key takeaways from the toy example, which has been moved Appendix A.4 "Toy example".

*** In Section 4 "WGAN-ALP" the metrics (Inception Score and FID) used in the evaluation are now described in more detail, as well as the reported numbers in Table 1 and how they were calculated. We also discuss how WGAN-LP fares in our implementation, and that monitoring the gradient norms during training shows that ALR in fact restricts the Lipschitz constant of the network.

*** Also in Section 4, motivated by the toy example, an additional experiment is described in which we add Batch Normalization to the critic in WGAN, and show that while it degrades performance when trained with GP, training with ALR is still successful. Details of the high-dimensional Progressive GAN example were also moved to this section from the Appendix, except the generated CelebA-HQ images which can be seen in Appendix A.5 "Images generated by Progressive GAN trained on CelebA-HQ".

---

> ### Author Response · Authors · 2019-11-15
> **Additional revision**
>
> *** Following the line of thought shared by AnonReviewer5, we have added a paragraph to Section 4 discussing the relationship of ALR and two-sided penalties.

---

### Decision · Program_Chairs · 2019-12-19

**Decision:**

Accept (Poster)

**Comment:**

This paper introduces an adversarial approach to enforcing a Lipschitz constraint on neural networks. The idea is intuitively appealing, and the paper is clear and well written. It's not clear from the experiments if this method outperforms competing approaches, but it is at least comparable, which means this is at the very least another useful tool in the toolbox. There was a lot of back-and-forth with the reviewers, mostly over the experiments and some other minor points. The reviewers feel like their concerns have all been addressed, and now agree on acceptance.